# Lived experiences and drivers of induced abortion among women in central Uganda

**Simon Peter Sebina Kibira** [1]*, **Melissa Stillman**[2], **Fredrick E. Makumbi**[1], **Margaret Giorgio**[2], **Sarah Nabukeera** [1], **Grace Kigozi Nalwoga**[3], **Elizabeth A. Sully** [2]

**1** School of Public Health, College of Health Sciences, Makerere University, Kampala, Uganda,
**2** Guttmacher Institute, New York, NY, United States of America, **3** Rakai Health Sciences Program, Rakai, Uganda

* pskibira@gmail.com, pskibira@musph.ac.ug

**Data Availability Statement:** All relevant data are within the paper and Supporting Information files.

**Funding:** The study on which this article is based was made possible by a grant from The David and

## Abstract

Although unsafe abortions are preventable, they are one of the leading causes of maternal mortality and morbidity. Despite the serious potential health consequences, there is limited published information about drivers and challenges of obtaining abortions in restrictive settings such as Uganda. This limits efforts to improve programing for preventing unsafe abortion and providing comprehensive post abortion care. This study sought to understand the drivers and explain the lived abortion experiences among women from central Uganda, in an effort to promote greater access to safe reproductive healthcare services, and reduce unsafe abortions. This qualitative study included 40 purposely selected women who self-reported an abortion, living in Kampala and greater Rakai district, Uganda. They were part of a larger survey using respondent driven sampling, where seed participants were recruited from selected facilities offering post-abortion care, or through social referrals. Data were collected from May to September 2021 through in-depth interviews. Audio data were transcribed, managed using Atlas.ti 9, and analyzed thematically. The findings show that the underlying drivers stemmed from partners who were unsupportive, denied responsibility, or had raped/defiled women. Career and education decisions, stigma and fear to disappoint family also contributed. Women had feelings of confusion, neglect, betrayal, or shame after conception. Abortion and post-abortion experiences were mixed with physical and emotional pain including stigma, even when the conditions for safe abortion in the guidelines were satisfied. Although most women sought care from health facilities judged to provide safe and quality care, there was barely any counselling in these venues. Confidantes and health providers informed the choice of abortion methods, although the cost ultimately mattered most. The mental health of women whose partners are unsupportive or who conceive unintendedly need consideration. Abortion provided psychological relief from more complicated consequences of having an unplanned birth for women.

Lucile Packard Foundation (2020-69738 to The Guttmacher Institute). The views expressed are those of the authors and do not necessarily reflect the positions and policies of the donors. The funders had no role in study design, data collection and analysis, decision to publish, or preparation of the manuscript.

**Competing interests:** The authors have declared that no competing interests exist.

## Introduction

Reproductive autonomy, or the ability for people to decide and control their own contraceptive use, pregnancy and childbearing, is an essential human right. However, this is not the case for many women, with global estimates indicating that nearly half (121 million) of pregnancies that occur annually are either mistimed or unwanted [1]. While evidence suggests that global rates of unintended pregnancy have declined over time, the proportion of unintended pregnancies ending in abortion has increased [2]. The most recent estimates suggest that more than half of unintended pregnancies end in induced abortion, and the global abortion rate between 2015 to 2019 was 39 abortions per 1,000 women of reproductive age (WRA) 15–49 years [2]. Abortion is a common health intervention that should be safe, if conducted using methods appropriate to the duration of the pregnancy as recommended by the World Health Organization by a person with the necessary skills [3]. Yet, nearly half of abortions that occur are unsafe [1, 4], with the largest burden occurring in the developing countries where it is highly restricted. Unsafe abortion remains a preventable, yet leading cause of maternal mortality and morbidity [5].

Despite the global decline in unintended pregnancies, the most recent national survey in Uganda indicated that the proportion of all most recent pregnancies/births considered unintended has increased to 46% (33% mistimed and 13% not wanted at all) in 2021 [6], from 41% in 2016 [7]. The problem of unintended pregnancy persists, partly due to a lack of access to sexual and reproductive health services and information, and limited use of any method of contraception. In Uganda, only 41.4% of all women aged 15–49 years are using a contraceptive method [8]. Contraceptive use rates have been rising at only about 1.2% per annum among all women since 2014 [6, 8–10], with high 12-month discontinuation rates [7, 11]. In 2021, 15% of all reproductive aged women and 26% of those in union were estimated to have an unmet need for family planning, meaning they did not want to become pregnant within the next two years but were not using any contraceptive method [6, 11].

A 2013 study estimated an abortion rate of 39 per 1,000 women of reproductive age (WRA) in Uganda [12], although recent modelling estimates have shown even higher figures (43 per 1000 WRA) [13]. The high nationally and sub populations rates of abortion [14] are not surprising given the rise in unintended pregnancy rates [11]. Abortion in Uganda is only allowed in specific cases such as severe maternal illness threatening the health of pregnant woman, HIV positive women requesting termination, rape, incest and defilement [15]. This is in line with the Maputo protocol that only requires States parties to take all appropriate measures to protect the reproductive rights of women by authorizing medical abortion in cases the above circumstances [16]. In Uganda post abortion care is not restricted and must be given to women irrespective of the cause or abortion status [15]. However, even though well provided under the African Union and National guidelines, the negative attitudes and deep social stigma [17–20], including among health workers [21] may complicate access to safe abortion and/or post abortion care services [19, 22, 23]. These issues need to be brought to light of policy makers, and program managers to drive up evidence informed decision making to minimize or eliminate unsafe abortions. Unsafe abortions are costly to Uganda's health system, with high expenditures on post abortion care [12, 21, 24], and deaths.

Given the highly restrictive nature of abortion in Uganda [21], there is limited information about the challenges that women inducing abortions face. This lack of evidence poses substantial challenges to provision and improvement of post abortion care programing. The purpose of this paper therefore was to understand the drivers of induced abortion and to explain the lived abortion experiences of women from central Uganda. We believe that lessons can inform the efforts to promote greater access to safe reproductive healthcare services, reduce unsafe

abortions, and ultimately contribute to reduction of maternal morbidity and mortality in Uganda and other abortion restrictive settings.

## Methods

### Ethics statement

The study was approved by Makerere University School of Public Health Research and Ethics Committee (protocol 921), the Guttmacher Institute's Institutional Review Board, and the Uganda National Council for Science and Technology (SS814ES). All participants provided written informed consent. Interviews were conducted in complete privacy—in spaces that the participants considered safe, in some cases away from their communities as preferred. A token of 20,000 UGX (USD 5.5) was provided to each participant as approved by the research and ethics committees. All interviews were conducted following an approved COVID 19 risk miti-gation plan, observing the guidelines provided by the Uganda Ministry of Health. SPSK, SN and GKN had access to personal identifying information of the participants during data collec-tion, but kept it confidential in a separate file away from the data transcripts and audio files.

### Design and sampling

A mixed-methods study was conducted in greater Rakai and Kampala city, in central Uganda in 2021. Kampala represented a purely urban setting, while greater Rakai represented a mix of rural and peri urban settings. The current paper is based on a qualitative approach to study lived abortion experiences among 40 women (19 in Kampala city and 21 in greater Rakai areas). These respondents were purposely selected from a respondent-driven sample (RDS) of 411 women whose eligibility criteria included: living in the study areas, having reported an abortion within the last 6 years from 2021, and aged 15–49 years at the time of the study. The RDS recruited respondents through identifying initial seeds from selected facilities offering post abortion care services in each site. Using a standard recruitment script, facility providers approached and asked women about their willingness to join the study. Willing women (potential seeds) were then provided with phone numbers of the study team, and they called in voluntarily, after leaving facilities. The providers did not influence recruitment. Recruited seeds were provided up to three coupons by the study team, with contact information to recruit other women from their social network whom they knew met the set criteria, and who they thought would voluntarily want to share their abortion information for this study. A writ-ten recruitment script was provided to the women to ensure use of standard procedures. All women were screened by the study team to ensure they met the criteria before admission into the study. A subset of these women, following the survey who were willing to share further information about their experiences were invited for follow up discussions after the survey to share their abortion experiences in-depth. A stratified purposive sample mixed with maximum variation of the quantitative sample characteristics was applied. We aimed to balance the two strata (study sites). However, following daily research team debriefings, we reached thematic saturation by the 18[th] respondent in Kampala, hence the slight imbalance between the two strata. Then, although not in equal numbers, the team also varied the sample by age, education levels, abortion methods used, sources, and experience with abortion (repeat and first-time abortion) to obtain views from a variety of experiences.

### Data collection

A team of four experienced female research assistants were trained to conduct the in-depth interviews. They were trained and supervised by SN, GKN, FM, and SPSK. The training

included a session about values clarification to minimize bias in collecting abortion data. All four had a minimum of a Bachelor's degree, certification in conducting research with human subjects, and had engaged in more than five studies before. The first interview for each interviewer was reviewed to improve the interview tools, especially for the probing questions. Regular virtual debrief meetings were held between the research assistants and the investigators (SPSK, SN, GKN, FM), to ensure smooth data collection. All in-depth interviews were audio recorded, with women's written consent. They lasted an average of 1hr and 12 mins. An in-depth interview guide was used with topics including how information regarding abortion is spread within social networks, factors related to their decision to abort, and their experiences with the most recent abortions throughout the process. Because participants were recruited from the survey respondents, data were collected for research purposes over several weeks; from 19th May to June 17th 2021, and then August 1st to September 1st 2021. There was an involuntary break between 18th June and 31st July 2021 due to the second wave of COVID-19 travel restrictions in the country.

## Data management and analysis

After each interview, audio files were submitted to the study coordinator for safe custody. The audios were transcribed verbatim and simultaneously translated into English, from Luganda, the local language. The transcripts were checked for completeness by the study coordinator and stripped of any identifying information, and audio files were deleted after confirming completeness. We used both deductive and inductive approaches to code the data; a sample of transcripts were read to develop a codebook, which was also informed by thematic areas from the interview guide. The transcripts were imported into Atlas.ti 9 for final coding using the draft codebook, allowing for further open coding. Two persons including SPSK were involved in the coding to minimize possible biases from a single view. Query reports and codes document tables were produced to aid the writing of the findings. The synthesized findings presented in this paper include excerpts of raw data as typical quotations. The paper is written following the consolidated criteria for reporting qualitative studies. A completed COREQ checklist is attached as supplementary file (S1 Table).

## Inclusivity in global research

Additional information regarding the ethical, cultural, and scientific considerations specific to inclusivity in global research is included in the S1 Checklist.

## Results

### Characteristics of participants

Table 1 summarizes the participants characteristics. The average age of the women was 28 years, with three quarters (75%) aged between 20 to 34 years. A third of women were in union (32.5%) while 42.5% had separated from partners. Majority had attained secondary or higher education level. Sixty percent had a child before the most recent abortion. Most (87.5%) women had the recent abortion from a health facility/clinic using medication abortion or surgical methods. For 70% of the women, this was their first abortion, and more than two thirds had the abortion within 2 years of the study. Most women had worked within the last 3 months.

Fig 1, summarizes the mapping of the findings from this study. It reflects what the 40 women reported to influence their decision to abort, experiences when they became pregnant before aborting, the sources of information during the period leading up to abortion, the

**Table 1. Characteristics of participants.**

| Characteristics | Number of women | Percent |
|---|---|---|
| **Age** | | |
| 15–19 | 4 | 10 |
| 20–24 | 11 | 27.5 |
| 25–34 | 19 | 47.5 |
| 35+ | 6 | 15 |
| **Marital status** | | |
| Never married | 10 | 25 |
| Currently in Union | 13 | 32.5 |
| Separated/Divorced | 17 | 42.5 |
| **Parity before recent abortion** | | |
| 0 | 16 | 40 |
| 1–2 | 16 | 40 |
| 3+ | 8 | 20 |
| **Education** | | |
| None or Primary | 16 | 40 |
| Secondary+ | 24 | 60 |
| **Repeat abortion** | | |
| Yes | 12 | 30 |
| No | 28 | 70 |
| **Method use for last abortion** | | |
| Medication | 17 | 42.5 |
| Surgical | 10 | 25 |
| Traditional methods (like herbs) | 13 | 32.5 |
| **Place of abortion** | | |
| Facility based | 35 | 87.5 |
| None facility based | 5 | 12.5 |
| **Timing of last abortion** | | |
| Less than 2yrs | 27 | 67.5 |
| 2 or more years | 13 | 32.5 |
| **Working status** | | |
| Worked in last 3 months | 31 | 77.5 |
| Not working | 9 | 22.5 |
| **Total** | 40 | 100 |

influences of their choices for abortion methods used and the places to go, and the challenges experienced during and after abortion.

## Drivers of abortion

**Partner and relationship-related drivers.** The main drivers of abortion were linked to unstable or challenging relationships and male partner's behavior exhibited soon after being informed about the pregnancy. Most women reported that the male partner was not supportive, when informed about the pregnancy. Worse still, 32.5% of the women reported that the men denied responsible for the pregnancy, or indicated that they were not interested (7.5%). This was reported by women in varied relationship types; cohabiting, long distance, and who had casual or one-off sexual relations. Some men abandoned their partners after receiving the pregnancy news, and moved to unknown places. Some were physically and psychologically

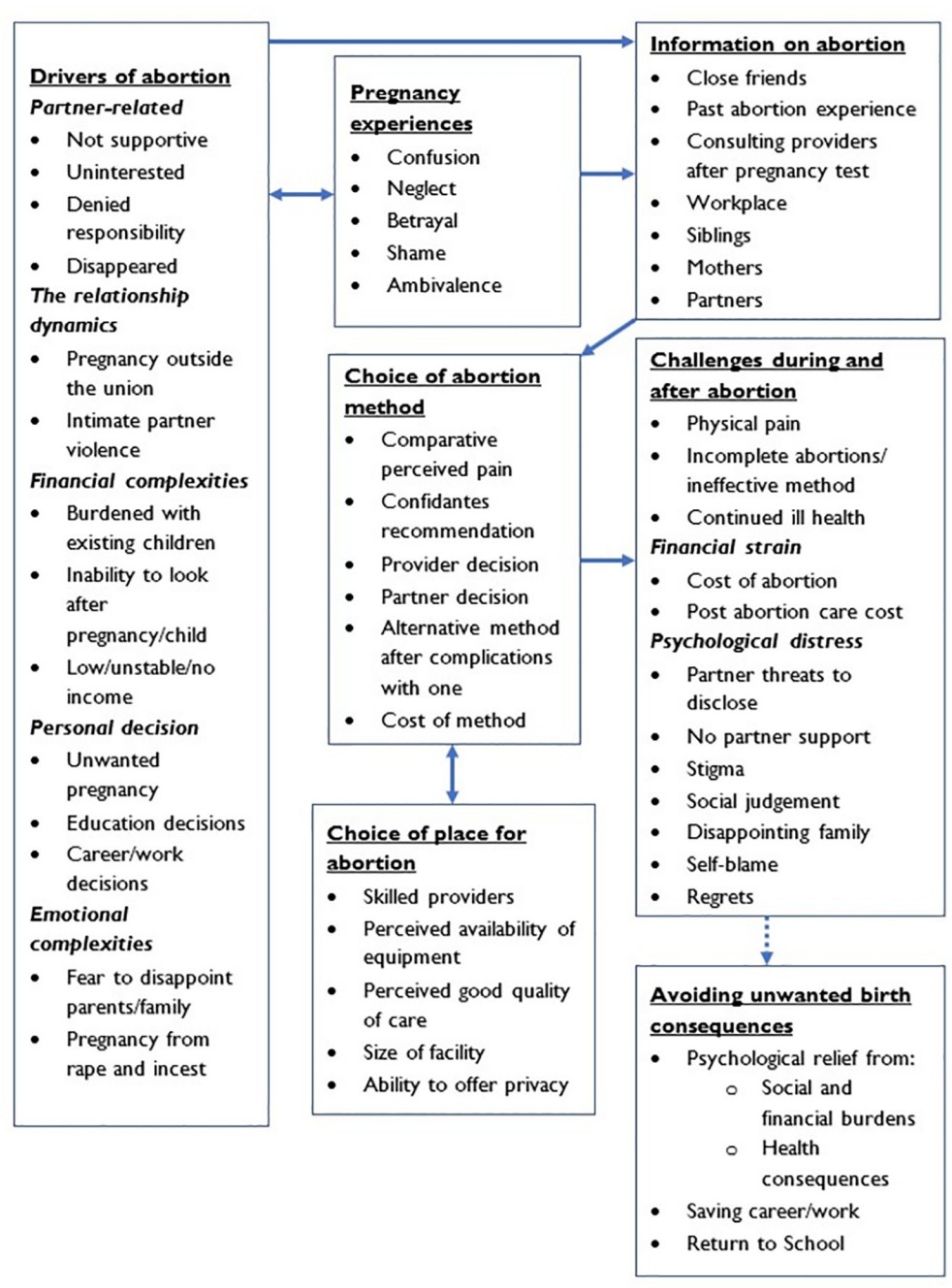

**Fig 1.** Mapping of experiences and drivers of abortion in greater Rakai and Kampala, Uganda.

abused in their relationships, and could not imagine conceiving in such circumstances. Even in cases where a few women became pregnant in relationships with several misunderstandings, these situations worsened when the pregnancy was announced.

*I conceived around September 2020 because by the time I aborted in February it was almost making five months. Honestly, I really did not want to because I conceived willingly. But*

*when I conceived we got serious misunderstandings with my husband. The situation was not good, the man was a teacher, from Mbale [Eastern Uganda]. When we had misunderstandings, it seems he had issues at his workplace too, so he left the job and went back to his birth place. He did not call, and when I called, he did not answer and eventually blocked my number. I then used my friend's phone to call him but whenever he heard my voice, he switched off. He left knowing I was two months pregnant because we did a pregnancy test at a clinic together. I imagined how to survive in such a condition. Then I thought of aborting when he left, but I thought that he will change his mind because there was no reason not to. . . I was terminated from work too while pregnant. So, I asked myself, "What I am I going to do with the pregnancy?" I don't know the home of the man or any of his relatives. Even if I call, he does not care. I was forced to abort.* **Participant 11_Rakai, age 25**

Unique to Rakai site, a third of the women there, reported that they and their confidantes aborted because the man responsible for the pregnancy was not their main partner at the time. They anticipated further complications in their unions after this. In one case the main partner had suspected he was not the father, and the respondent's choice was abortion. One woman was still in a grieving period; just days after losing a partner, and got pregnant with a brother-in-law who offered 'support' during this time. It was complex for her to report such a pregnancy immediately after the husband's death.

*I missed my periods then I started feeling feverish, so I went and bought the pregnancy test kit, then I tested myself only to find I was pregnant. I informed my husband. Then he said, 'that is impossible. I am not responsible for that pregnancy.' The fact was that he was not responsible, I had got it outside of marriage. . .I knew where I had got that pregnancy, I thought it would look weird, maybe I would continue with the pregnancy and give the child to my husband. But already he had denied it, so I decided to have an abortion.* **Participant 20_Rakai, age 39**

**Financial drivers.** Where women reported the partners to be irresponsible, the financial strain and inability to care for the pregnancy and later the child, complicated the situation, in both study sites. The women without paid work or with meagre erratic incomes from casual work, could not imagine taking care of the pregnancy. Some, both young and older women, were already strained with the burden of having other young children.

*The truth is that I did not have enough money to care for it [pregnancy/child], so I decided to abort. But, before then, my heart would say 'please leave it', yet it continued to grow and the man's [responsible partner] manners continued to worsen. Actually, he eventually switched off his phone.* **Participant 11_Rakai, age_25**

**Personal decision drivers.** A quarter of all women stated that they had an abortion simply because they did not want the pregnancy at the time. Nine women were still students at the time they got pregnant, and abortion was the only way to continue their education.

*Sincerely I did not want the pregnancy right away, because I did not wish to get pregnant I think I would have strangled the baby myself after birth [if she had not successfully aborted]* **Participant 3_Rakai, age 18**

Other reasons included the fear of disappointing parents or guardians among young people still living with their guardians, a stressful work environment that was not ideal for pregnancy and fear of losing their job as a source of income, among those engaged in work. Some got pregnant from rape by family-friends or employers, or through incest; one woman conceived by the uncle.

*My father had his best friend who used to come home often, and even stay over. I used to prepare food for them, and mum had already left [separated]. One-day he came home. When I finished washing and prepared food, I went to take a bath, but I found him in the sitting room after. I went to the bedroom to dress up and he followed me. He raped me from the bedroom and left. I was still young [on 14 years]. . . So, after three months I started feeling bad. I was feeling weak, lost appetite, I wanted to sleep all the time. . .It was in November when I learnt I was pregnant. I was in the compound then felt headache, felt dizzy, I collapsed. I gained consciousness while in the school sickbay. Then I asked the nurse what had happened to me. She asked if I had ever had sex. She told me I was pregnant. . . I was feeling ashamed. . . I wasn't fat at that time, I was small but tall. I could not imagine people seeing me pregnant at 14 years, then they say that girl misbehaved at that age.* **Participant 7_Rakai, age 16**

## Feeling after confirming pregnancy

Women were asked how they felt after confirming the pregnancy prior to their most recent abortion. Many felt bad and confused, because the pregnancy was not intended. Some reported initially feeling positive, but after losing the support of the partners or significant others, the feelings towards the pregnancy changed. These women reported that they would have carried the pregnancy if such care was guaranteed. There were also feelings of shame and self-blame, especially in cases of conception outside the union or after rape/incest. A few women were ambivalent of the next steps at the time when they learnt about the pregnancy. One woman reported feeling suicidal tendencies at the time.

*Before testing I was not feeling well and when I confirmed the feeling became worse. I said to myself that I have no job, I have no permanent place of residence. I came [to the city] to look for money, I have a school going child, I don't know what to do. I tried calling him [partner] so that we would meet and talk. . . When we met, he told me that "I am not interested in giving birth at the moment". I asked him what I should. His response was; 'you know better.' I went back thinking that it was a joke or the response was out of fear. I called again, he never picked my calls, I called several times without any response. What would I have done? Someone was housing me with my child. I had just got my job. My next step was to abort. When I consulted my friends they also advised to abort. I didn't intentionally do it and that is why I told you that if you have somewhere to stay, please settle and give birth to your children. It is not good to abort. . . I was patient thinking that he would change his mind, I waited and waited but I was becoming weak. I could vomit everything I ate, getting out of the house was hard for me yet I had to work.* **Participant 2_Kampala, age 42**

## Choosing the appropriate location/provider for an abortion

Half of the women reported that they got information about where to go through their social networks. Women learned from their friends who had used the same facilities before, or already knew where to obtain the abortion prior to getting pregnant; through knowing someone who worked there or (for one women) had worked in the same facility themselves.

*This friend of mine was already a friend to him [health worker]. I wonder whether she had aborted before. She was taking there many people. The health worker was well known for*

*conducting abortions, they always have their phone numbers. Actually, she just called and told him I would like to bring my sister. . .So, she directed me and escorted me too. . .. I wasn't aware of any other place. I didn't know anywhere.* **Participant 4_Rakai, age 28**

A third of the women sought information about where to obtain an abortion on their own in order to maintain privacy and prevent information leaking in their community. Some went to health facilities for a pregnancy test after missing their menstrual period, and while there, they discussed abortion as an option, and received referrals to place that could provide. Others searched for information without stating that it was them who needed to abort.

*Okay that one [traditional healer] was my friend because she knew us from childhood. I could hear about her as an old woman who usually gives out herbal medicine. So, I thought that because she is a herbalist, she might be knowing these medicines to use. That is why I went to her. I deceived her that it was my friend who needed the herbs.* **Participant 11_Rakai, age 25**

A few found out where they could abort from through family, including their sisters, and although rarely reported, through the mother and/or partner.

*When I confirmed to him [partner] that I agreed to abort, I asked him where I should go. He directed me to the health center. . .There is a health worker that he had connected me to. I went to him and he told me to lie on the bed and he inserted the pills.* **Participant 8_Kampala, age 33**

In choosing where to obtain their abortions, nearly half of the women prioritized places with skilled health care workers and available equipment. Skilled healthcare workers were indicative of good care, for some women. Although the availability of equipment was not always known, women used indicators like the size of the facility to assume they had equipment. Over a third of women chose a place based on their judgement of the ability to offer privacy. This was either through offering abortion together with general services, or because they were not frequented by people, and/or far enough from their community to minimize any recognition. Privacy also included the environment in the procedure room; women chose places where there was visual and auditory privacy. The cost of a place was also used to choose places.

*What enticed me was the availability of a doctor, very professional and he also works in known hospitals. I selected it because when things fail at his facility, he could easily refer me to the next level. I also went there for privacy, I wanted to be treated from a health facility and come out well without my partner at home knowing what had transpired. All I wanted was to be in the hands of a trained personnel. . . I went to two clinics; the first one was asking for 100,000UGX, the second one my friend who had accompanied me was known there, so I told them that I didn't have money and they reduced the cost.* **Participant 2_Kampala, age 42**

## Choice of methods

Over two thirds of the women reported procuring abortion services from the health facilities using medication abortion or surgical procedures. For medication abortion, they obtained what they described as "tablets" from pharmacies on their own, at clinics and inserted by the providers, or brought to their home by the confidantes. None of the respondents specified the brand names of the tablets. Surgical methods were obtained at clinics, but most women could only describe the process and appearance/ shape of equipment used.

For some, their method choice was based on previous experiences that was more painful, thinking they would have less pain with a new method; from surgical to medication. Others were advised by their confidantes on the 'best' method based on what they knew. A few women only went to seek abortion services at facilities or (in one case) were taken by a partner and did not know what method to use, but relied on the provider's choice.

*I refused surgical procedures because it is what I used for my first abortion and it was so painful. This time they explained to me that there was a method that was not painful. He [male partner] had also told me about it. . . . They used pills, there are tablets that they inserted in me and the rest I put them under the tongue. I went back home. They told me go back home and sleep. It will come out. I thought that I would just feel it coming out but it was so painful. Eventually it came out.* **Participant 8_ Kampala, age 33**

*We went in and I was injected on my thigh and then terminated it [pregnancy]. . . They just brought something like a tube and they inserted it there, you feel it inside here while outside here he is sucking / pulling it out. The pain was too much, it was so painful because I had never done it before.* **Participant 9_ Rakai, age 18**

*I lay on the bed then he came in with some equipment on a tray. I didn't see what they were. He told me to widen my legs then inserted some equipment and I felt as if he was pumping something. . . I felt everything he was doing; the pain was too much. I held his arm. He told me 'do not hold my arm, do you want me to spoil an organ here?' Let me to do what am supposed to do.' But I was in pain.* **Participant 12_ Rakai, age 34**

Other women chose traditional methods including herbal abortifacients. These were known through referrals to traditional providers by confidantes who had used them. There were also several women who simultaneously used modern (medication) and the herbs for abortion. Others used the herbs to help with the post abortion effects including managing pain and what they referred to as "cleansing the uterus" after abortion. The most used herb for cleansing the uterus was "kamunye" [Hoslundia opposita]. Others included *commelina africana*, and *phytolacca dodecandra*. Among users of herbal abortifacients, there were reported complications that needed further assistance from the health centers, clinics, or pharmacies.

*I knew about them [herbal plants], but I didn't know where to get them from. I knew there was herbal medicine but the specific type was chosen by her called 'luwoko' [phytolacca dodecandra] and she brought it for me, but we know these things. There's another herb called 'ennanda' [commelina africana] which we insert in our private parts, to terminate a pregnancy.* **Participant 1_Kampala, age 37**

*I shared with my friend and she told me about a woman who gives herbs. I took them and by 6pm abdominal pains had started, then after it came out. I saw that something had come out so I quickly to cut it and it fell down then I got up, padded and stayed there for a week. But when I came back, I noticed some things remained inside although I was walking. . . I went to hospital and told them that I had a miscarriage but there are things that remained inside.* **Participant 5_Rakai, age 38**

## Challenges experienced during and after abortion

Respondents reported physical, financial, emotional, and psychological challenges during and after the abortion process. These included physical pain and lethargy, experience of stigma, financial strain, and psychological distress. Some reported using ineffective methods resulting

in incomplete abortions that negatively impacted an already challenging situation. Even those who did not directly experience these consequences of abortion, several expressed fear of stigma, social judgment, exposure of their secrets by partners, and disappointing the family.

**Physical challenges.** Most women, regardless of age, location and abortion methods used, reported experiencing extreme pain during the abortion process and the period directly after. There were varied coping mechanisms following the onset of pain including swallowing pain killers to medicinal herbs.

*I experienced very bad side effects. If my friend did not come early in the morning, the effects might have worsened. . .I do not understand these tablets [for medication abortion] because I experienced terrible abdominal pains at night from about 2am, and I thought I was going to die inside the house. But, I had already swallowed them and there was nothing to do. In the morning, I started bleeding very much and I got worried again that I was going to die inside the house, which would be shameful. Then the foetus came out but there was a problem within my uterus, which I did not know about. It was later that I started experiencing more pain. I consulted my friend and she brought 'Kamunye' [Hoslundia opposita] to drink. . .However, I realized that tummy was swelling and I told my friend that 'you know what let me go to hospital because there appears something that might not be right.' At the hospital, I was told that if the swelling had increased beyond, I would have died. I had clots in various places and so if I had not gone to the hospital, the medical officer told me I would have died because many women have died like that.* **Participant 7_Kampala, age 31**

**Financial challenges.** Women reported financial strain with some not being able to afford their preferred methods of abortion. Beyond abortion services, there were costs related to managing the post abortion care, like buying sanitary pads, pain killers and post abortion care services at facilities, in case of incomplete abortion.

*I was afraid of them [tablets] but my money would afford that because the other method was costlier. They were going to suck out the fetus and it cost 100,000/ = yet I had only 50,000shs.'* **Participant 17_Kampala, age 24**

Several women relied on partners who promised to provide financial support but backtracked, forcing them to seek alternative support from friends or relatives.

*It was a clinic. I told him [health worker] that I didn't have the money and he asked 'how much can you afford?' I told him 30,000/ =. He laughed and said 'I can help you do it at 50,000/ = but with no extra medicine given'. So I had to look for the 50,000/ = which took time to get because I was not working. It is my friend and my niece who contributed to get the 50,000/ =, which we paid to the provider to conduct the abortion, but he didn't give me any tablet. After the abortion, he said 'you go and whatever happens do not tell me.' We went home but I had too much pain the whole night then the fetus came out in the morning, it was about three months old because we took over a month to get the money. The bleeding was too much, I would take boiled herbs, I was so dizzy. My friend would come and buy me millet flour, sometimes cassava flour for porridge, until God helped me to recover.* **Participant 12_Rakai, age 34**

**Psychological and emotional challenges.** Women also reported suffering psychologically for a long period after the abortion. A few younger women and those aborting for the first time, as well as the few who conceived from rape and incest, faced challenging psychological

distress. Misperceptions about the consequences of abortion also worsened the psychological distress.

> *I cried and regretted that I will never do it again. I wished I had given birth. The pain worsened every time I thought about it. During conferences, they advised us not to abort because it is a bad omen and one may never conceive again. Whenever I listened to that, I felt guilty, it would hurt. . . I got scared that I might fail to conceive and I made a promise to only get married after I become pregnant. I wanted to get married after confirming that I can conceive, but I failed my own promise, because I ended up aborting again. Another thing was seeing my breasts sagging while other girls were firm. I felt bad losing my beauty yet I could not tell any of the girls that I aborted.* **Participant 9_Kampala, age 25**

Although rarely reported, there was fear for stigmatization due to abortion, with some young women experiencing it in the community. They hid from community members, friends, family, and housemates, endeavoring to conceal evidence that could breed suspicion. They feared to disappoint family, especially parents. One woman feared to report the history of abortion while seeking care for her current one, because of perceived judgment from a provider.

> *A certain gentleman in our village told me that, 'you are deceiving, you are not sick, you are not suffering from fever, you say the truth that you aborted, someone can tell, you are not like that usually.' They told me that I had an abortion.* **Participant 15_Rakai, age 18**

There were women who used ineffective methods resulting in complications and at times incomplete abortions. Some women, at the time of the study still suffered what they thought were the effects from the methods used. They also reported that they knew other women who experienced similar consequences. These psychologically affected their quality of life.

> *I used herbs and eventually had a foul smell from my private parts, which took time to clear. . . After I got a discharge, it doesn't smell nor itch but I have it [to date] and I cannot disclose to people, I keep quiet and suffer in silence.* **Participant 18_Rakai, age 33**

For a few women, the process of aborting at the health facilities/clinics, was daunting. The fear of what was going to happen and the equipment to be used, which they had not heard about or seen, coupled with a lack of counselling complicated some experiences.

> *He said I am going to use some equipment, but you must be strong, then I was wondering how the equipment work, I was in great fear. They [tools] can really scare. Have you ever had tooth extraction? The chairs in the dentist room! So, when I got to this room there was a certain smell, the chairs had a funny shape, you feel as if you are in hell, you feel 'yes, it is time to die'. The health worker mixed medicine, he was preparing his machines, I was all shivering. . . and by the time he attended to me, honestly, I was feeling the [perceived] pain.* **Participant 19_Rakai, age 25**

## Abortion helped avoid worse consequences

Although women reported enormous challenges in the process, for most, abortion provided a sense of relief from the expected consequences of having an unplanned birth. Most said they would have given birth but expected a more complicated life. For those still in school, it would spell the end of their education, and those with informal jobs, the end of their work.

*Schooling would have stopped at that and I think I would not have sat for senior six. I would have ended there but the child would be alive.* **Participant 6_ Kampala, age 25**

The most feared consequence was the financial strain resulting from having a pregnancy and a (another) child, yet many were already living in hard financial situations. Abortion provided an option to circumvent these potentially worse problems. Statements like '*no one would have supported me,*' '*I would be back in the village suffering*' or '*I would have lost my job*' were common.

*I don't know what would have happened because I was finished. I was going to hustle with that pregnancy yet I had no help. I already have two other children here and they are still young and yet I am the father and I am the mother at the same time. I would be pregnant without any help. Let me not lie to you, it would have been so bad. . . when I aborted and returned in my normal state, I began working for these other [three] children, because I am the mother and the father..* **Participant 3, Kampala_age 28**

*Of course, I would be suffering. . .because you do not have the Child's father, and even the first child is in the same state (fatherless). . .You won't be able to buy a mattress, you won't buy smearing oil, you won't buy clothe because all the children are your responsibility. How do you think you would feel?* **Participant 21, Rakai_age 38**

There were also health and other social consequences anticipated in case no abortion was done. Women living with HIV could not imagine the additional burden of pregnancy, while fending for other children. A few adolescent girls and young women imagined the impact on their future marital relationships if they had a child, and those whose partners were unsupportive imagined no one who would have cared for them.

*I can't regret, the termination helped me so much because whenever someone is pregnant there are changes in terms of health yet I have to work, I have to work. . . I don't know but it would have been a very difficult financial and health situation. Health wise I have to take ARVs, then the pregnancy, so health wise too, it wouldn't have been easy. These two conditions and the stress too yet I have to take care of the family.* **Participant 12, Rakai_age 34**

*A lot would happen it would be the end of the world. I was small I think I would lose more weight. Sincerely you face a lot of challenges; where to stay, in case they chase you away from home, where do you go? Who could accommodate me? The nurse [that was helping her] was also married with children and renting. The world can come to an end before me and you die or commit suicide.* **Participant 7, Rakai_age 16**

## Discussion

This study provides an exploration of abortion drivers and lived experiences of women obtaining abortion in central Uganda. For most women, the underlying drivers were partner related; including being involved with partners who were either financially or emotionally unsupportive, and/or who denied responsibility for the pregnancy. For some, the pregnancy was a result of violence, rape, defilement or incest. Such women did not want to have a child in such a relationship, but if circumstances were favorable, they would have carried the pregnancy. Some women reported terminating their pregnancy in order to salvage a career or education path; some also expressed that the shame and stigma around their pregnancies as well as the fear of disappointing family after conceiving contributed. Women expressed negative experiences after confirming pregnancy, with feelings of confusion, neglect, betrayal by loved ones, shame after rape or incest or pregnancy from infidelity. Abortion and post abortion experiences were

mixed with physical and emotional pain in addition to realized or feared stigma. Most women sought abortion care from formal facilities that they deemed safe and able to provide quality care, with skilled providers and equipment available. Choices of the method and place were based on recommendations by confidantes, and affordability, but providers also played a role in method choice while at the facility. However, very few women reported any pre or post abortion counselling; an essential component of safe abortion care. Lacking supportive information and counseling services, women did not know what to expect from the abortion process and reported pain and confusion around their experiences. Many, based on misinformation, were concerned about the longer-term consequences of their abortions. Amidst these complex experiences however, many women reported feeling psychologically relieved when they hypothesized what the social and financial situation would have been if they had a (another) child.

Although the main drivers of abortions are expected to stem directly from unintended pregnancies [21], as was the case with a quarter of the women in this study, the reasons women have abortions are more nuanced. A common theme in our study was that for most women, the underlying drivers of the decision came from external factors, often partner related. Similar findings regarding partner influence have been reported before [25, 26]. Men play an important role in women's health decisions [23] especially in patriarchal settings like Uganda with limited women's empowerment and/or reproductive autonomy. One's paternity determines a sense of belonging in Uganda because naming of a child follows the father's line. Carrying a pregnancy in cases where the man denies responsibility or is untraceable is daunting and has many ramifications for women. This tradition makes it complex and may result in stigma for the child born with no defined father figure and therefore no clear lineage. Abortion as seen in this study may be the 'best' option for a woman to counter the lasting effects of such denials. While most women cited partner-related issues in their decision to abort, they did not rely on their partners for support; rather, they turned to their confidantes for help or chose to navigate the experience alone. With unsupportive partners, the financial complexities played a big role in the abortion decision, as evidenced elsewhere [27]. Costs like transport and distance impact on abortion decisions and place where to, even in more developed settings [28]. The cost of abortion and post abortion care for many was financially taxing in this study. Some could not afford the preferred methods, and others sought support of friends and/or relatives, in cases of uncooperative partners. This has also been reported in another study in Kampala and Mbarara, Uganda [19].

Our study also shows that stigma and shame surrounding unintended pregnancy drive abortions. Young women especially expressed fear to disappoint their family when they became pregnant. The expectations placed on children as the future of the family in many cases is well known. Thus, any disappointment resulting from pregnancy has far reaching effects, including the shame that the family especially in rural well-connected communities may experience. Another study in Ghana among adolescents indicated similar social influence [29]. A few women in our study had experienced rape and/or incest and could not explain this to the family and community, or live with the baby that resulted from such an experience. Sexual abuse like defilement in which the culprit is a family member (incest) is secretly handled in many settings to minimize shame on the family. In such cases, the victims suffer silently to 'protect' the family name and seek abortion with minimal support. Women expressed confusion, feeling of neglect, betrayal by loved ones, and shame after pregnancy from rape, incest or infidelity. Others were uncertain about next steps. With negative attitudes rife in the community, several women were uncertain about what the family or community would think about their decision. The pre and post abortion periods were lonely or with only close confidantes. Although not explicitly reported in this study, the heavy religious prohibitions of abortion

[30], in all circumstances, as part of the tradition, may also have shaped the pre and post abortion experiences. Previous studies have reported secrecy as part of abortion [26] and society judgement from religious leaders expecting women to carry pregnancies, even in conditions like rape [30], defilement or incest.

Some women in the Rakai site aborted because they had conceived with another man outside the union and were scared of the consequences. An earlier study in central and western Uganda revealed restrictive abortion attitudes, with men believing a woman can only abort where conception is not from the main partner [19]. Women in this study sometimes sought abortion clandestinely even when they had conceived through rape or incest, or tested positive for HIV while pregnant. Yet, all these would have qualified for safe and legal abortion services under the national guidelines [15].

In this study, most women sought care from the formal health facilities using medication abortion or surgical methods. This shows that even in settings where most abortions are carried out clandestinely because of restrictions, women can and do access safer abortion services. Most women seeking care in health facilities looked out for skilled providers expected to provide quality care, assumed availability of equipment to support the abortion, and to offer privacy. The choice of what method to use was based on mainly recommendation by confidantes in the social networks who had aborted.

Past experiences also counted for those who had repeat abortions. The role of the provider was brought to light in this study, with some women noting the method used was decided by the provider. However, ultimately the cost mattered, given that many had financial constraints; some choosing a place or method that wasn't the first choice. Although rarely reported, traditional methods like herbal abortifacients were used. Several women also reported trying to use both a modern (often medication abortion) method in combination with herbal. Herbs were also used to manage post abortion complications or "cleansing the uterus" after. In an earlier study in Uganda, Prada et al [31], also reported several herbs as well as dangerous objects were used in abortion, especially among rural women.

Challenges during and after abortion were enormous for women in this study. Most reported excruciating pain in the process and some because of incomplete abortions using ineffective methods. Women experienced psychological distress as a result of the abortion experience and in some cases partners' threats to disclose their abortion. The post abortion stigma and social judgement [20] and the disappointment of the family was often feared, living in worry of being discovered and deciding to isolate or be less social. This supports the view that although abortion may be a private matter, it can have social ramifications as indicated in another study among men in western Uganda [32]. Self-blame and lasting regrets even after successful abortions were rife, some still wishing they had not aborted. The lack of counselling in health facilities before and after abortion could have complicated the experiences. Given the limited awareness and ambiguity in implementing abortion care guidelines, abortions tend to be clandestine in nature. Providers and women may therefore aim to quicken the process, in fear, further worsening the experiences. Although Uganda is signatory to the Maputo protocol where States parties are required to take all appropriate measures to provide adequate, affordable and accessible health services, including information to women especially those in rural areas [16], there is a clear gap as evidenced in this study.

Although the abortion experience was challenging for all women in the study, it is worth noting that many felt psychologically relieved when they imagined what the consequences would have been if they kept the pregnancy and had a(another) child. Relief from further financial strain, the ability to look after the children they already have, and the ability to continue with their education was paramount. However, this relief only came after having to navigate an undesirable and difficult abortion experience, in most cases without much support.

The strength of this study is that we obtained lived experiences from women across two varied settings, a city and a peri-urban location. The study participants were purposely selected from a large diverse survey sample of women with a recent abortion. This provided maximum variation of views from multiple demographic and social backgrounds. The study has some limitations. Because we started recruitment of seeds in the RDS at post abortion care facilities, this may have resulted in having more women in the sample receiving facility-based abortions. All women were recruited from RDS, so the views in our qualitative data does not include women who did not disclose their abortions to others, and who may be more isolated in their experiences.

## Implications

This study has a number of implications for policy and programming. There is need to raise awareness about the reasons for abortion. Abortion- and pregnancy-related stigma leads women to seek abortion clandestinely, which means some women seek abortion from untrained providers or use ineffective or unsafe methods (even under conditions where abortion should be legally provided for). Stigma also adds to the psychological distress many women experience throughout their abortion, so addressing abortion stigma is central to improving women's abortion and post abortion experiences.

The awareness of National abortion guidelines among women seeking abortion and healthcare providers, including under what conditions and where women can access legal services needs to be given attention. There should also be community awareness on what to expect from an abortion, how to recognize post abortion complications, where and when to seek treatment.

There should be emphasis on counseling training as part of post abortion and safe abortion care training, including sharing information to combat misperceptions about the consequences of abortion.

## Conclusions

Partner relations are the most impactful factors contributing to abortion among women in this study; when they do not provide financial and emotional support after conception, deny responsibility or leave women to bear the brunt alone. Rape, defilement or incest should not be underestimated as a cause of unintended pregnancies and resulting into abortion, with culprits going unpunished. The stigma and fear to disappoint family after conceiving and the self-blame even in circumstances that women have no control over, like rape, also contributes to unsafe abortion.

The confidantes in women's social network play a crucial role recommending methods and places to seek abortion care. With limited awareness in their network about guidelines for legal abortion, decision to seek abortion and post abortion care may also be impacted.

The abortion and post abortion experiences are unpleasant in restrictive settings, even when conditions for safe abortion are satisfied. The mental health of women whose partners are unsupportive or who conceive unintendedly needs consideration. Yet, there was barely any pre and post abortion counselling. Although undesirable, abortion provided psychological relief from consequences of bearing an unintended child in complex situations.

## Supporting information

**S1 Checklist. Inclusivity in global research.**
(DOCX)

**S1 Text. Excerpts from the transcripts.**
(DOCX)

**S1 Table. Consolidated criteria for reporting qualitative studies checklist.**
(DOCX)

## Acknowledgments

The authors are grateful for the support received from partners at Rakai Health Sciences Program and Reproductive Health Uganda; Joseph Kagaayi, Kenneth Buyinza and Edward Kiggundu. We also thank the research assistants who collected the data; Resty Nakayima, Prossie Aliwebwa, Sheila Kisakye, and Margaret Nansubuga. We are indebted to the women who shared their personal sensitive and touching stories.

## Author Contributions

**Conceptualization:** Simon Peter Sebina Kibira, Fredrick E. Makumbi, Margaret Giorgio, Sarah Nabukeera, Elizabeth A. Sully.

**Data curation:** Simon Peter Sebina Kibira, Melissa Stillman, Margaret Giorgio, Sarah Nabukeera, Grace Kigozi Nalwoga, Elizabeth A. Sully.

**Formal analysis:** Simon Peter Sebina Kibira, Melissa Stillman.

**Investigation:** Simon Peter Sebina Kibira, Fredrick E. Makumbi, Margaret Giorgio, Sarah Nabukeera, Grace Kigozi Nalwoga, Elizabeth A. Sully.

**Methodology:** Simon Peter Sebina Kibira, Melissa Stillman, Fredrick E. Makumbi, Margaret Giorgio, Sarah Nabukeera, Grace Kigozi Nalwoga, Elizabeth A. Sully.

**Project administration:** Simon Peter Sebina Kibira, Fredrick E. Makumbi, Sarah Nabukeera.

**Resources:** Elizabeth A. Sully.

**Software:** Simon Peter Sebina Kibira.

**Supervision:** Simon Peter Sebina Kibira, Fredrick E. Makumbi, Sarah Nabukeera.

**Validation:** Melissa Stillman, Fredrick E. Makumbi, Margaret Giorgio, Elizabeth A. Sully.

**Visualization:** Simon Peter Sebina Kibira.

**Writing – original draft:** Simon Peter Sebina Kibira.

**Writing – review & editing:** Melissa Stillman, Fredrick E. Makumbi, Margaret Giorgio, Sarah Nabukeera, Grace Kigozi Nalwoga, Elizabeth A. Sully.

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
