## [Decision Letter · Decision Letter 0]

22 Jun 2023

PGPH-D-23-00643

Lived experiences and drivers of induced abortion among women in central Uganda

Dear Dr. Kibira,

Thank you for submitting your manuscript to PLOS Global Public Health. After careful consideration, we feel that it has merit but does not fully meet PLOS Global Public Health’s publication criteria as it currently stands. Therefore, we invite you to submit a revised version of the manuscript that addresses the points raised during the review process.

Thank you for submitting your article to PLOS Global Public Health. Your manuscript is being decisioned as a major revision. Kindly incorporate the comments that reviewer #2 especially has suggested to strengthen the argument and clarify certain points. I have highlighted them for your review.

1) "...the authors insist on the restricted legal environment for abortions in Uganda, without mentioning the state of application of the Maputo Protocol, even though Uganda has been the 28th signatory of this protocol since 2010, which is dedicated to the promotion of maternal reproductive health, in particular liberalization and access to safe abortion. It is, therefore, necessary to succinctly demonstrate the existing gap and the barriers to the complete application of this protocol according to the country's context."

2) "....should provide the necessary explanations in the technique of sampling to justify such a repartition between the greater Rakai (21 respondents) and the city of Kampala (19 respondents) when Kampala is far more populated than Rakai".

3) "...data need to be presented better. The percentage gives a rough estimate of the coverage of the variable in the total sample. The author should therefore add a column for percentage in Table 1".

4) "The partner's participation, perception, and opinion are critical in explaining and interpreting abortion drivers. It is desirable to know to what extent the respondents expressed themselves on the support or not of the partner in relation not only to the announcement of the pregnancy but especially to the decision to abort in the participants' general characteristics "

5) "The extract from the interview chosen by the author on financial drivers does not correspond to the theme but rather shows rejection on the partner's part after the pregnancy announcement, without emphasizing the financial problem as a motivation or consequence of the abortion. The same applies to the extract chosen for the personal drivers, where the participant, although still at school, seems to have been somewhat influenced by her partner, the teacher".

6) "The author failed to effectively demonstrate how the community drivers, particularly religion, culture, and tradition, influence the issue of abortion in the result. These factors play a significant role in shaping attitudes toward the topic. Clarification is needed."

We look forward to receiving your revised manuscript.

Kind regards,

Sreeparna Chattopadhyay, Phd

Academic Editor

Journal Requirements:

2. We have noticed that you have uploaded Supporting Information files, but you have not included a list of legends. Please add a full list of legends for your Supporting Information files after the references list. 

Additional Editor Comments (if provided):

Regards,

Dr Sreeparna Chattopadhyay on behalf of PLOS Global Public Health

Reviewers' comments:

Reviewer's Responses to Questions

**Comments to the Author**

1. Does this manuscript meet PLOS Global Public Health’s publication criteria? Is the manuscript technically sound, and do the data support the conclusions? The manuscript must describe methodologically and ethically rigorous research with conclusions that are appropriately drawn based on the data presented.

Reviewer #1: Yes

Reviewer #2: Yes

2. Has the statistical analysis been performed appropriately and rigorously?

Reviewer #1: N/A

Reviewer #2: N/A

3. Have the authors made all data underlying the findings in their manuscript fully available (please refer to the Data Availability Statement at the start of the manuscript PDF file)?

Reviewer #1: Yes

Reviewer #2: Yes

4. Is the manuscript presented in an intelligible fashion and written in standard English?

Reviewer #1: Yes

Reviewer #2: Yes

5. Review Comments to the Author

Reviewer #1: This is a very pertinent study areas and a cause for avoidable morbidity and mortality in low-and middle-income countries. The study has unearthed various drivers for induced abortions, and provide practical policy direction to address this global public health issue. However, in line 281, the spelling for "could" should be corrected. In addition, the authors should include possible areas for future studies.

Reviewer #2: After careful and thorough analysis, this manuscript is admissible, as it generally meets the requirements of our Journal (PLOS). However, the author must make a few amendments before it is fully admissible.

-In his argument, the author insists on the restricted legal environment for abortions in Uganda, without mentioning the state of application of the Maputo Protocol, even though Uganda has been the 28th signatory of this protocol since 2010, which is dedicated to the promotion of maternal reproductive health, in particular liberalization and access to safe abortion. It is, therefore, necessary to succinctly demonstrate the existing gap and the barriers to the complete application of this protocol according to the country's context.

- The author should provide the necessary explanations in the technique of sampling to justify such a repartition between the greater Rakai (21 respondents) and the city of Kampala (19 respondents) when Kampala is far more populated than Rakai.

-The data need to be presented better. The percentage gives a rough estimate of the coverage of the variable in the total sample. The author should therefore add a column for percentage in Table 1.

-The partner's participation, perception, and opinion are critical in explaining and interpreting abortion drivers. It is desirable to know to what extent the respondents expressed themselves on the support or not of the partner in relation not only to the announcement of the pregnancy but especially to the decision to abort in the participants' general characteristics (table 1).

-The extract from the interview chosen by the author on financial drivers does not correspond to the theme but rather shows rejection on the partner's part after the pregnancy announcement, without emphasizing the financial problem as a motivation or consequence of the abortion. The same applies to the extract chosen for the personal drivers, where the participant, although still at school, seems to have been somewhat influenced by her partner, the teacher.

-The author failed to effectively demonstrate how the community drivers, particularly religion, culture, and tradition, influence the issue of abortion in the result. These factors play a significant role in shaping attitudes toward the topic.

Clarification is needed.

6. PLOS authors have the option to publish the peer review history of their article (what does this mean?). If published, this will include your full peer review and any attached files.

**Do you want your identity to be public for this peer review?** For information about this choice, including consent withdrawal, please see our Privacy Policy.

Reviewer #1: **Yes: **

Reviewer #2: No

---

## [Editor Report · Decision Letter 1]

8 Aug 2023

PGPH-D-23-00643R1

Lived experiences and drivers of induced abortion among women in central Uganda

Dear Dr. Kibira,

Thank you for submitting your manuscript to PLOS Global Public Health. After careful consideration, we feel that it has merit but does not fully meet PLOS Global Public Health’s publication criteria as it currently stands. Therefore, we invite you to submit a revised version of the manuscript that addresses the points raised during the review process.

We look forward to receiving your revised manuscript.

Kind regards,

Sreeparna Chattopadhyay, Phd

Academic Editor

Journal Requirements:

Additional Editor Comments (if provided):

Dear Authors,

Thank you for revising this manuscript. While you have taken most of the comments on board from the reviewers, I picked up on some areas that need addressing. Please see below:

1) The qualitative description [25] on 108 which the paper is *bases* had phenomenological [26] overtones, and included 40 women (19 109 in Kampala city and 21 in greater Rakai areas [based not bases]. Also the term phenomenological overtones needs clarification. Phenomenology is a methodology. What do you mean here?

2) You have mentioned a team of 4 RAs collected all the data - are they included as authors? Without their labour, this work would have been impossible. I would urge you to expand the author list to include them in the manuscript.

There are extensive copyediting and grammatical issues with this manuscript. Please make sure it is grammatically correct before submitting the final version. I have given some examples below but it is not my role to be a copyeditor. I am an editor.

3) "Most women reported that their male partner was vividly not caring or supportive when informed" Please rephrase, clunky sentence.

4) " One woman was still in a grieving period, just days after losing their partner, and got pregnant with a brother-in-law who offered ‘support’ during this time." Why their since it refers to a woman?

5) "Other reasons included the fear of disappointing parents among young people still living with their caregivers’, a stressful work environment that was not ideal for pregnancy and fear of losing their job as a source of income, among those engaged in work." Why caregivers? And why an apostrophe after caregivers? Shouldn't it be parents or families? Caregivers is used as a term for people who are sick or with disabilities and the reference doesn't seem to suggest that is the case.

6) "Although the availability of equipment was not always known, they used indicators like the size of the facility to denote assume they had equipment. " Denote, assume? which one?
---

## [Editor Report · Decision Letter 2]

31 Aug 2023

PGPH-D-23-00643R2

Lived experiences and drivers of induced abortion among women in central Uganda

Dear Dr. Kibira,

Thank you for submitting your manuscript to PLOS Global Public Health. After careful consideration, we feel that it has merit but does not fully meet PLOS Global Public Health’s publication criteria as it currently stands. Therefore, we invite you to submit a revised version of the manuscript that addresses the points raised during the review process.

Please submit your revised manuscript by . If you will need more time than this to complete your revisions, please reply to this message or contact the journal office at globalpubhealth@plos.org. Please include the following items when submitting your revised manuscript:

We look forward to receiving your revised manuscript.

Kind regards,

Sreeparna Chattopadhyay, Phd

Academic Editor

Journal Requirements:

Additional Editor Comments (if provided):

You have mentioned that the study uses phenomenological design. The characteristics of phenomenological research includes a deeper engagement with perceptions, memories, affect and imagination and a sample size of 10-15. Of these your study only uses perceptions and experiences. In-depth interviews of the format conducted here doesn't automatically qualify it for a phenomenological approach. The way in which this data is presented is a straightforward qualitative approach that uses thematic analysis. You need to alter the language and the terms to reflect that this is the case in this study. 

Also the quotes should be followed by a description or interpretation. This interpretive approach is completely skipped in some parts and included in other. Please rectify this.

Challenges expressed during and after abortion: While the rest of the themes have sub-themes nested within it, this is one long section without themes. Please break this up into sub-themes that come up in the analysis. 

The discussion section is well-written.
---

## [Editor Report · Decision Letter 3]

27 Oct 2023

Lived experiences and drivers of induced abortion among women in central Uganda

PGPH-D-23-00643R3

Dear Dr. Kibira,

We are pleased to inform you that your manuscript 'Lived experiences and drivers of induced abortion among women in central Uganda' has been provisionally accepted for publication in PLOS Global Public Health.

Best regards,

Sreeparna Chattopadhyay, Phd

Academic Editor